# The Association of Maternal Accompaniment at Family Dinners and Depressive Symptoms of Korean Adolescents

**DOI:** 10.3390/ijerph17051743

**Published:** 2020-03-07

**Authors:** Jae-Young Lee, Seul Lee, Eun-Cheol Park, Juyeong Kim, Sung-In Jang

**Affiliations:** 1Medical Courses, Yonsei University College of Medicine, Seoul 03180, Korea; bigjenny@naver.com (J.-Y.L.); kathy0318@naver.com (S.L.); 2Institute of Health Services Research, Yonsei University, Seoul 03180, Korea; ECPARK@YUHS.AC; 3Department of Preventive Medicine & Institute of Health Services Research, Yonsei University College of Medicine, Seoul 120-752, Korea; 4Department of Health and Human Performance, Sahmyook University, Seoul 01795, Korea

**Keywords:** depressive symptoms, family dinners, adolescents, mental health

## Abstract

This study aims to investigate the association between family dinners involving the participation of both mother and her adolescent child and depressive symptoms within the adolescents. Data from 2183 mother–child pairs obtained from the Korea National Health and Nutrition Examination Survey IV–VI (2010–2013, and 2015) were employed in the analysis. The dependent variable of this study was depressive symptom of adolescents. Maternal accompaniment at family dinners was the variable of interest. Logistic regression analysis was performed to calculate odds ratios (ORs) and 95% confidence intervals (CI) to analyze the association between family dinners including both mother and adolescent and depressive symptoms within the adolescent. According to the results, maternal absence in family dinners was significantly associated with higher odds of depressive symptoms in adolescents (OR = 1.42, 95%CI: 1.01–1.99). In particular, the association was strong among adolescents aged 12–15, female adolescents, those with mothers without depressive symptoms, and city dwellers. This study showed that maternal absence at family dinners was strongly associated with depressive symptoms of adolescents. For the sake of adolescent mental health, it may be necessary to consider the implementation of policies that emphasize the importance of maternal accompaniment at family dinners and encourage the presence of mothers at the dinner table.

## 1. Introduction

Depressive symptoms are prevalent in modern society that some consider it a “mental cold” that anyone can experience at least once in their lifetime [1]. According to studies by Ma and Cho and Kim et al., approximately 8.7% of adolescents in the Republic of Korea are identified as depressed [2,3]. The high rates of depressive symptoms in Korean adolescents underscore the importance of investigating the factors related to adolescent depression.

In adolescents, depression is associated with an increased risk of developing anxiety disorders (e.g., panic disorder) and various behavioral disorders (e.g., learning difficulties, hyperactivity). It also has a negative impact on the overall development of young adolescents by increasing the risk of conflict in social relations. Additionally, depressive disorders have a high recurrence rate and are difficult to treat once they become chronic [4] Worse still, the correlation between depression and suicide is particularly strong in adolescents, especially when the depressed emotions persist [5]. For these reasons, we sought to investigate the factors related to depressive symptoms in Korean adolescents.

Multiple factors are known to be associated with depressive symptoms in adolescents such as personal or family history of depression, major life changes, trauma, or stress, and certain physical illnesses. Family-related factors have long been emphasized as important for healthy emotional development in adolescents [6,7,8,9,10]. In particular, being part of a “healthy family”—that is, a family that engages in communication and activities together to enable sharing of thoughts and values and perceiving affection during mild interactions—plays a positive role in adolescent development [11]. One way of encouraging natural communication and interaction between family members is through family meals. Consequently, researchers have been focusing more and more upon the relevance of family meals in recent years.

Studies are underway around the world to examine the relationship between family dinners and an adolescent’s depressive symptoms. Most studies show that the frequency of family dinners is positively correlated to emotional health, prosocial behavior, and life satisfaction [12]. In South Korea, there have been numerous studies on the effect of dining with family on the nutritional and physical development of adolescents [13]; however, few have examined how family dinners affect adolescent psychological development. One study in 2009 investigated the relationship between the frequency of family meals and life satisfaction and psychological state. This study found that a higher frequency of family meals is associated with higher life satisfaction and healthier psychological state [14]. However, this previous study concentrated on life satisfaction and overall psychological state, not depression. Additionally, the study did not control for important factors known to influence psychological state (including depression), such as family income and the mother’s marital and employment status. Therefore, our study concentrated on adolescent depression, examining whether family meals influence an adolescents’ depressive symptoms even after controlling for the influence of factors such as family income and the mother’s employment and marital status. 

Previous studies on the effects of family dinners concentrated on whether the whole family participated. However, since South Korean mothers generally assume the bulk of childcare duties, we focused on the mother’s presence at an adolescents’ dinner. Furthermore, as the incidence of depression is sensitive to the sex of the adolescent, we evaluated the results according to the adolescent’s sex [15]. Also, since we used data from adolescents of a wide age range (12–19), we were able to categorize them into younger adolescents of ages 12–15 and older adolescents of ages 16 or above. Since each age group has different characteristics [16], examining the disparity in results regarding the ages of the adolescents is necessary.

We used data from the whole of South Korea, not just from metropolitan areas. As a result, we also analyzed differences between urban and rural areas. Finally, we examined differences in the results according to whether or not mothers experienced depressive symptoms themselves, as maternal depression is known to have a significant effect on an adolescent’s risk of developing depression [4]. Taking the factors mentioned above into consideration, this study aims to analyze the association between maternal accompaniment at family dinners and the depressive symptoms of adolescents.

## 2. Materials and Methods

### 2.1. Data

We employed data from the Korea National Health and Nutrition Examination Survey (KNHANES) from 2010–2012 and 2013–2015 (i.e., the KNHANES V and VI, respectively). The KNHANES has been conducted since 1998 by the Korea Centers for Disease Control and Prevention (KCDC), and it utilizes a multistage clustered probability design to select representative samples stratified by geographic location, sex, and age. All KNHANES participants provided written informed consent, and voluntarily participated in the survey. The KNHANES are open to the public. Anybody who submits a suitable form can download the KNHANES data via website [17]. Subjects of the KNHANES were randomly selected as a household to complete questionnaires about their health and nutritional information. The questionnaires specifically asked for information about participants’ health, nutritional information, socioeconomic status, and anthropometric measurements. The completed questionnaires were reviewed and entered into the database by trained staff.

### 2.2. Participants

A total of 48,445 people participated in the KNHANES 2010–2012 (KNHANES V) KNHANES 2013, 2015 (KNHANES VI). Data from the KNHANES 2014 was not employed since the data regarding depressive symptoms of the mothers, which is an important variable that could influence the depressive symptoms of the adolescents, were not collected. From this pool, we chose households consisting of “children with an identified mother” and “mother with identified children.” We then selected adolescents age 12–19 and mothers age 30 and older—19,720 people in total. By merging the data of mothers and adolescents, participants could be grouped into 3096 pairs of mother and child in the same household. Because we aimed to focus on whether the mother and adolescent had dinner together, we excluded mothers (n = 197) and adolescents (n = 250) that had missing data on participation in family dinners. We also excluded adolescents that had missing values for depressive symptoms (n = 297), dietary conditions (n=1), limitation of activity (n = 4), and regular walking (n = 1). We also excluded mothers with missing data for marital status (n = 3), household income level (n = 18), education level (n = 119), working condition (n = 6), and alcohol consumption (n = 17). Finally, 2183 adolescents (matched with 2183 mothers) were included in the analysis.

### 2.3. Measures

We collected information on household income; adolescents’ age, sex, depression, dietary condition, frequency of walking, behavioral restrictions, and subjective body cognitions; and mother’s economic status, weekly working hours, education status, depression, income, marital status, frequency of alcohol consumption, and behavioral restrictions. 

We assessed whether the mother and her adolescent child had dinner together using their answers to the question, “During the past year, did you usually have dinner with someone else?” and the sub-question “Who did you have dinner with?” The response options for this sub-question were “family members” and “someone other than a family member.” If both mother and child gave an affirmative answer to the first question and answered that they had dinner with their family members, then we considered that the mother and adolescent had dinner together. To assess depressive symptoms in both the mother and adolescent, we used their answers to the following question: “For the past year, did you ever feel sad or hopeless constantly for two weeks, severe enough to disrupt your social life?” The response options were “yes” and “no.” Responses to this question have been used as the dependent variable in previous research on depression [18].

The adolescents’ demographic factors such as their age (12–15; 16 and older) and sex (male; female) were taken into consideration. The adolescents’ dietary condition was categorized as “ample amounts of various foods,” “ample amount of food without much variety,” “sometimes do not have enough food,” and “often do not have enough food.” We combined these last two categories into a new group: “insufficient amount of food.” Subjective body cognitions were classified into five groups: “severely skinny,” “slightly thin,” “average,” “slightly obese,” and “severely obese.” We reclassified these categories into “underweight,” “average,” and “overweight.” To assess limitation of activity in both the mother and adolescent, we used their answers to the following question: “Are you currently restricted to your daily life and social activities due to health problems or physical or mental disorders?” We counted affirmative answers as “limited activity” and negative responses as “activity not limited.” The adolescents’ regular walking exercise was assessed based on the frequency of walking, and it was assessed as the number of days walking more than 10 minutes per week, with response options of 0 to 7 days. We reclassified their answers into yes/no.

For maternal demographic factors, maternal age (30–40; 41–45; 46 and older), education level (middle school graduation or less; high school graduation; college graduation or more), household income level (low, middle, and high), and region (city; rural) were used. Marriage status of the mother was assessed by the answer to the question, “What is your current marital status?”. The answers were classified into four groups: “married and living together,” “married and living separately,” “separation by death,” and “divorce.” Maternal work related factors such as work schedule (day work, night work, shift work and others, and unemployed), and working hours (h) were included. Maternal current alcohol drinking status was classified into “yes” or “ no,” and year was included.

### 2.4. Statistical Analysis

The general characteristics of the study population were described using the means and standard deviations (S.D.) or frequencies with percentages. The t-test and chi-square test was used to compare differences in various characteristics according to whether the mother and her children had dinner together (Table 1). A logistic regression analysis was utilized to calculate odds ratios (ORs) and 95% confidence intervals (CI) in order to analyze the associations between family dinners involving the mother and adolescent and the depressive symptoms of the adolescent, while controlling for the potential confounding variables (Table 2). Additionally, we performed subgroup analyses of the associations according to the adolescents’ sex, adolescents’ age, region, and whether or not the mother had depressive symptoms (Table 3). Two-tailed p-values were considered statistically significant when less than 0.05. All statistical analyses were performed using SAS ver. 9.4 (SAS Institute, Inc., Cary, NC, USA).

## 3. Results

Participants’ general characteristics are shown in Table 1. Among the adolescents, the prevalence of depressive symptoms was 8.94% (n = 164). Approximately 50.76% of mothers and adolescents ate dinner together with their family (n = 931). Among these mothers and adolescents, 7.52% (n = 70) of the adolescents experienced depressive symptoms. Among the mothers and adolescents that did not eat dinner together, the prevalence of adolescent depression was 10.41% (n = 94). 

Table 2 shows the adjusted ORs of depression in adolescents according to whether the mother and adolescent ate dinner together with their family, after controlling for household income; adolescents’ age, sex, dietary condition, regular walking exercise, limitation of activity, subjective body cognitions; and mother’ age, education level, depressive symptoms, marital status, region, work schedule, working hours, current alcohol drinking status, and limitation of activity. We found that the odds of the adolescent having depression were higher among mothers and adolescents that did not eat a family dinner together compared to those who did eat a family dinner together (OR = 1.42, 95% CI: 1.01–1.99).

Table 3 shows adjusted ORs of depression in adolescents according to whether mothers ate dinner with their children, stratified by adolescent’s sex, residential area, and whether the mother has depression. For the adolescent’s sex stratification, we found that the odds of a girl having depression were higher when her mother was not present at dinner than when her mother was present at dinner (OR = 1.814, 95% CI: 1.085–3.032). There was no significant difference between the odds of experiencing depressive symptoms for adolescent males who ate with their mother and those who did not (OR = 1.243, 95% CI: 0.763–2.025). Regarding region, adolescents who lived in a city had higher odds of suffering depressive symptoms when their mother was absent at family dinners (OR = 1.724, 95% CI: 1.167–2.548), while the odds of adolescents who live in rural areas were not significantly different. For mothers with depression, we found higher odds of depression in adolescents whose mother had no depressive symptoms and did not eat dinner with them (OR=1.92, 95% CI = 1.277–2.888), while adolescents with mothers who had depressive symptoms showed no significant values. Finally, the odds of younger adolescents (ages 12–15) having depression were higher when the mother did not eat dinner with them than when they did (OR=1.609, 95% CI: 1.055–2.452), while the older ones (ages 16 and above) did not show significant difference. 

## 4. Discussion

The primary objective of this study was to examine the association between family dinners in which both mother and adolescent child were present, and the depressive symptoms of the adolescent. The results of this study show that the rate of depressive symptoms in adolescents is 8.0%, which is in line with previous studies that showed that the rate of depressive symptoms in adolescents to be around 8.7% [2,3]. The study results showed that the absence of the mother during dinner was significantly associated with higher odds of depressive symptoms in adolescents. In particular, the association between maternal accompaniment at family dinners and an adolescents’ depressive symptoms was most strongly prevalent among adolescents aged 12-15, female adolescents, those with mother’s without depressive symptoms, and city dwellers.

Our findings are consistent with previous studies that assess the influence of family dinner on children’s mental health. Elgar et al. demonstrated that the frequency of family dinners is negatively correlated to internalizing and externalizing symptoms and positively correlated to emotional wellbeing, prosocial behavior, and life satisfaction [12]. According to Musick et al., family dinners may protect children from depression and risky behaviors by providing a regular and comforting context to check in with parents about their day-to-day activities and to connect with them emotionally [19].

Our findings, which showed an association between a mother’s absence at family dinners and higher odds of depressive symptoms for her adolescent child, can be explained by possible mechanisms. Links between family dinners and mental health are partially attributable to the ease of communication between adolescents and parents [12]. Levin and Currie examined the association between mother–child communication and life satisfaction among youth, and found that the ease with which they can communicate with their mothers serves as a protective factor for mental wellbeing [20]. In addition, mother–child communication was found to be a protective factor for the mental status of adolescents, even during stress conditions such as cancer and death of a father [21]. A mothers’ warm, sensitive, and engaged communication is associated with lower levels of maladaptive grief and depressive symptoms in her children [21]. As adolescents mature, they generally tend to become oriented toward their peers than their mothers, which leads to the decrease of time for mother–child interaction [22]. Family dinner is proper way to provide adolescents with chances for mother–child conversation, maternal advice giving, and discussions of emotions [23]. Therefore, it is understandable that maternal accompaniment in family dinners have a strong correlation with depressive symptoms in adolescents. 

In addition, this study found that the association between maternal accompaniment at family dinners and adolescents’ depressive symptoms was strong among girls. This finding agrees with that of a study showing that girls who do not communicate with their mothers show lower life satisfaction than boys who do not communicate with mothers, or even girls who do not communicate with fathers [20]. Also, the association was strong among the adolescents aged 12-15. According to Hadiwijaya et al., as adolescents get older, hormonal changes act as the main force driving them to separate themselves from their parents to become autonomous and independent individuals [24]. It can be assumed that the importance of maternal accompaniment fades as the adolescents mature and perceive mental independence from their mothers. Another point is that the association was strong among the urban adolescents. Cities are known to negatively influence the mental health of city dwellers [25]. In fact, the risks of severe mood swings, anxiety disorders [26], and schizophrenia [27] are notably higher among people living in cities. It is possible that since adolescents who live in cities are exposed to various urban stress factors, it is more important for them to relieve stress, which is closely correlated with depression, via maternal interaction [28]. Also, the association was strong among the adolescents whose mothers accompany them at family dinners when their mothers are not depressed. The association between a mother’s presence at dinner and adolescents’ depressive symptoms was not significant when mothers had depressive symptoms. Mother–child communication between mothers who do not have depressive symptoms and her adolescent children can work as a protective factor for the mental status of adolescents [21]. However, according to Corona et al., depressed mothers had poorer parenting quality, including low warmth, high hostility, and high conflict, which is related to increased risk of depressive symptoms in adolescents [29]. Therefore, it can be assumed that the positive correlations between mother–child interactions over dinner and the mental health of adolescents mostly applies when the mother is not depressed. 

There are several political implications for our findings. In order to decrease the possibility of adolescent depression, government policies that emphasize the importance of maternal attendance at family dinners should be reinforced. Such policies should help not only mothers, but society as a whole to acknowledge the importance of maternal attendance at family dinners for adolescent mental health. Mothers, especially those with younger adolescents around the ages 12 to 15, should be encouraged to have dinner with her children through government programs and projects. It is necessary that new policies create an environment where mothers can provide emotional support and engage in daily conversation with her adolescent children through accompanying them at family dinners. 

## 5. Strengths and Limitations of This Study

The current research has some limitations. Paternal factors were not considered in this study, as we wanted to focus on the maternal influence. Furthermore, many adolescents in the KNHANES dataset, especially those raised by single mothers, had missing values for the father’s identification numbers, which prevented us from considering paternal factors. While maternal care is known to have a stronger influence on an adolescent’s development [30], paternal care and its related factors are still important. Another issue is that we did not consider whether family dinners took place at home. Family dinners include the act of preparing, cooking, eating, and cleaning up together as a family [31]. Eating out, which typically involves only eating, was also counted as a “family dinner” in this study. Additionally, the frequency of family dinners is not provided in the KNHANES data, so our results are based on whether the participants had family dinner or not. Also, the important variables of adolescent depression and maternal depression were only divided into two categorical variables, “yes/no.” A deeper examination of the main variable, as well as a more thorough analysis, would have been possible if depression scales for adolescents and their mothers had been used. To add, the study utilized cross-sectional data, which impairs our ability to determine a causal association between maternal present at family dinners and depressive symptoms among Korean adolescents. The fact that many mother–child pairs were excluded due to the lack of data on considered variables could also be a limitation. Additionally, this study did provide evidence that maternal attendance at family dinners and depressive symptoms in adolescents are strongly associated, but whether maternal absences cause depressive symptoms in children requires further study. Finally, the observational nature of our study leaves room for residual confounding and other potential sources of bias.

The current study also has some important strengths. First, this study analyzed the direct relationships between maternal factors and their adolescent children’s depressive symptoms using nationally representative data. Also, we examined mother–child interactions and maternal influences by focusing specifically on family dinners in which both the mother and the adolescent participated, rather than simply focusing on the existence or frequency of family dinners, as in other studies [32,33]. We considered not only the child-related factors of adolescent depression, but also the maternal factors, such as the mother’s depressive symptoms, education level, household income, marital status, region, work schedule, working hour, current alcohol drinking status, and limitation of activities. Rather few studies on family dinners have adjusted for as many maternal factors as ours. Furthermore, many previous studies were conducted among families living in large cities, making it easier to gather survey data in urban areas. However, the current study used reliable data on South Korea as a whole.

## 6. Conclusion

The findings of this study suggest that maternal accompaniment in family dinners and depressive symptoms in adolescents are strongly associated. In particular, the association between maternal participation in family dinners and an adolescent’s depressive symptoms was most prevalent with those adolescents aged 12-15, female adolescents, those with mothers without depressive symptoms, and city dwellers. In order to decrease the possibility of adolescent depression, it might be necessary to create policies and projects, as well as a social atmosphere, that encourage mothers to participate in family dinners for her adolescent children.

## Figures and Tables

**Table 1 ijerph-17-01743-t001:** General characteristics of adolescents aged 12–19 years and their mothers by depressive symptoms of the adolescents.

Variable	N	%	Depressive Symptoms				*p*-Value
			No		Yes		
			N	%	N	%	
	2183	100.0	2009	92.0	174	8.0	
Family dinner							0.042
Not accompanied by mother/child	1043	47.8	947	90.8	96	9.2	
Accompanied by mother/child	1140	52.2	1062	93.2	78	6.8	
Offspring Factors							
Adolescent’s age (Mean ± S.D.)	14.5 ± 1.9		14.4 ± 1.9		14.6 ± 1.9		0.233
12–15	1490	68.3	1374	92.2	116	7.8	0.639
16 or above	693	31.8	635	91.6	58	8.4	
Adolescent’s sex							
Male	1154	52.9	1071	92.8	83	7.2	0.155
Female	1029	47.1	938	91.2	91	8.8	
Limitation of activity							
Limited activity	58	2.7	50	86.2	8	13.8	0.097
Activity not limited	2125	97.3	1959	92.2	166	7.8	
Subjective body cognition							
Underweight	547	25.1	508	92.9	39	7.1	0.187
Normal	943	43.2	874	92.7	69	7.3	
Overweight	693	31.8	627	90.5	66	9.5	
Dietary condition							
Ample amount of various food	969	44.4	888	91.6	81	8.4	0.814
Ample amount of food without much variety	1142	52.3	1054	92.3	88	7.7	
Insufficient amount of food	72	3.3	67	93.1	5	6.9	
Regular walking exercise							
Yes	2022	92.6	1859	91.94	163	8.06	0.580
No	161	7.4	150	93.17	11	6.83	
Maternal Factors							
Maternal Age (Mean ± S.D.)	43.2 ± 4.2		43.2 ± 4.1		42.9 ± 4.5		0.397
30–40	572	26.2	520	90.9	52	9.1	0.476
41–45	1044	47.8	967	92.6	77	7.4	
46 and older	567	26.0	522	92.1	45	7.9	
Level of education							
Middle school graduation or less	172	7.9	154	89.5	18	10.5	0.444
High school graduation	1222	56.0	1126	92.1	96	7.9	
College graduation or more	789	36.1	729	92.4	60	7.6	
Income							
Low	180	8.3	162	90.0	18	10.0	0.535
Middle	1242	56.9	1143	92.0	99	8.0	
High	761	34.9	704	92.5	57	7.5	
Marriage status							
Married and living together	2043	93.6	1883	92.2	160	7.8	0.807
Married and living separately	9	0.4	8	88.9	1	11.1	
Separation by death	35	1.6	32	91.4	3	8.6	
Divorce	96	4.4	86	89.6	10	10.4	
Region							
City	1853	84.9	1701	91.8	152	8.2	0.343
Rural	330	15.1	308	93.3	22	6.7	
Work schedule							
Day work	1303	59.7	1201	92.2	102	7.8	0.639
Night work	174	8.0	156	89.7	18	10.3	
Shift work and others	65	3.0	61	93.9	4	6.2	
Unemployed	641	29.4	591	92.2	50	7.8	
Working hours (Mean ± S.D.)	27.2 ± 22.5		27.1 ± 22.4		27.9 ± 23.5		0.687
Maternal depression							
Yes	273	12.5	236	86.5	37	13.6	0.000
No	1910	87.5	1773	92.8	137	7.2	
Current alcohol drinking status							
Yes	1479	67.8	1371	92.7	108	7.3	0.095
No	704	32.3	638	90.6	66	9.4	
Limitation of activity							
Limited activity	127	5.8	119	93.7	8	6.3	0.474
Activity not limited	2056	94.2	1890	91.9	166	8.1	
Year							
2010	486	22.3	448	92.2	38	7.8	0.593
2011	467	21.4	423	90.6	44	9.4	
2012	435	19.9	405	93.1	30	6.9	
2013	446	20.4	408	91.5	38	8.5	
2015	349	16.0	325	93.1	24	6.9	

Notes, S.D, standard deviation.

**Table 2 ijerph-17-01743-t002:** Adjusted odds ratios for the association between maternal accompaniment at family dinners and depressive symptoms of adolescent.

Variable	Adolescent Depressive Symptoms			*p*-Value
	aOR ^†^	95% CI		
Family dinner				
Not accompanied by mother/child	1.447	1.02	2.05	0.037
Accompanied by mother/child	1.000			
Offspring Factors				
Adolescent’s age				
12–15	1.053	0.73	1.53	0.786
16 or above	1.000			
Adolescent’s sex				
Male	0.804	0.58	1.11	0.182
Female	1.000			
Limitation of activity				
Limited activity	1.929	0.87	4.27	0.105
Activity not limited	1.000			
Subjective body cognition				
Underweight	0.805	0.53	1.23	0.307
Normal	0.761	0.53	1.09	
Overweight	1.000			
Dietary condition				
Ample amount of various food	1.625	0.58	4.60	0.567
Ample amount of food without much variety	1.438	0.52	4.01	
Insufficient amount of food	1.000			
Regular walking exercise				
Yes	1.000			
No	0.906	0.47	1.73	0.766
Maternal Factors				
Maternal Age				
30–40	1.316	0.82	2.11	0.438
41–45	1.054	0.70	1.59	
46 and older				
Level of education				
Middle school graduation or less	1.283	0.68	2.43	0.745
High school graduation	1.050	0.73	1.50	
College graduation or more				
Household income level				
Low	1.158	0.58	2.31	0.916
Middle	1.039	0.72	1.50	
High				
Marriage status				
Married and living together	0.772	0.35	1.69	0.874
Married and living separately	1.094	0.12	9.99	
Separation by death	0.613	0.15	2.50	
Divorce				
Region				
City	1.290	0.80	2.09	0.299
Rural				
Work schedule				
Day work	0.955	0.54	1.69	0.736
Night work	1.231	0.62	2.46	
Shift work and others	0.692	0.21	2.24	
Unemployed				
Working hours	1.000	0.99	1.01	0.987
Maternal depression				
Yes	1.000			
No	0.479	0.32	0.72	0.001
Current alcohol drinking status				
Yes	0.814	0.58	1.13	0.222
No	1.000			
Limitation of activity				
Limited activity	0.548	0.25	1.21	0.135
Activity not limited	1.000			
Year				
2010	1.037	0.60	1.80	0.586
2011	1.358	0.80	2.30	
2012	0.977	0.55	1.73	
2013	1.259	0.73	2.17	
2015	1.000			

Notes: aOR, adjusted odds ratio, Cl, confidence interval. †Adjusted for offspring’s factors (age, sex, limitation of activity, subjective body cognition, and regular walking exercise) and maternal factors (age, level of education, household income level, marriage status, region, work schedule, working hours, depression, and limitation of activity), and year.

**Table 3 ijerph-17-01743-t003:** Result of subgroup analysis of the association between maternal accompaniment at family dinners and depressive symptoms by adolescent’s sex, adolescent’s age, maternal depressive symptoms, and region.

Variable		Adolescent Depressive Symptoms				
		aOR^†^	95% CI		*p*-Value	Col %^a^
Offspring Factors						
Adolescent’s age						
12~15	Not accompanied by mother/child	1.66	1.11	2.47	0.014	29.9
	Accompanied by mother/child	1.00				36.8
16 or above	Not accompanied by mother/child	1.15	0.58	2.29	0.682	25.3
	Accompanied by mother/child	1.00				8.0
Adolescent’s sex						
Male	Not accompanied by mother/child	1.03	0.62	1.71	0.907	24.7
	Accompanied by mother/child	1.00				23.0
Female	Not accompanied by mother/child	2.07	1.26	3.39	0.004	30.5
	Accompanied by mother/child	1.00				21.8
Maternal Factors						
Maternal depression						
Yes	Not accompanied by mother/child	0.40	0.16	1.00	0.050	8.6
	Accompanied by mother/child	1.00				12.6
No	Not accompanied by mother/child	2.00	1.35	2.95	0.001	46.6
	Accompanied by mother/child	1.00				32.2
Region						
City	Not accompanied by mother/child	1.62	1.11	2.35	0.012	50.6
	Accompanied by mother/child	1.00				36.8
Rural	Not accompanied by mother/child	0.54	0.17	1.66	0.278	4.6
	Accompanied by mother/child	1.00				8.0

Notes, aORa adjusted odds ratio; CI: confidence interval; ^†^ Stratified Adjusted OR by adolescent’s sex, adolescent’s age, mother’s depressive symptoms, and region was calculated by adjusting offspring’s factors (age, sex, limitation of activity, subjective body cognition, and regular walking exercise) and maternal factors (age, level of education, household income level, marriage status, region, work schedule, working hours, depression, and limitation of activity), and year after stratifying by adolescent’s sex, adolescent’s age, mother’s depressive symptoms, and region. ^a^ Column percent (%) based on 174 adolescents the total number of participants with depressive symptoms.

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
