# Peer review of "The Association of Maternal Accompaniment at Family Dinners and Depressive Symptoms of Korean Adolescents"

_ijerph, 2020, doi:10.3390/ijerph17051743_

Round 1
Reviewer 1 Report
The advantage of the paper is the extensive collected research material covering 1,834 adolescents and their mothers. The tables were also prepared in an extremely careful manner. However, I would like to point out a number of shortcomings in the submitted manuscript. First, you should: indicate the aim of the paper more clearly, complete the description of research tools, thoroughly improve the discussion, including the part describing the strengths and weaknesses of the paper. When discussing the limitations of the study, it would be worthwhile to address the following comments.
Main reservations:
- Introduction should end with a clearly defined aim and research questions. Research questions should be consistent with the results presented sequentially and give direction to the discussion.
- The main result variable identifying children and mothers with depression symptoms is a very simple tool – a single question with Yes / No answer categories. As far as the wording of the question is acceptable, simplified categories of answers seem to be questionable. This should be highlighted as a weakness of the study. The percentage of adolescents with depression symptoms is low. It has not been compared with other studies.
- "Behavioural restrictions" turned out to be an important variable. This part of the questionnaire was not described, nor was this significant correlation referred to in the discussion.
- The age factor has not been considered, and more frequent depressive symptoms can be expected in older adolescents. It has not been examined how the influence of the mother accompaniment in family dinners (and other mother related variables) changes with the child's age.
- Table 3 should be supplemented with a column with the percentage of children with depression symptoms. The reader may assume (looking only at OR values) that, for example, depression symptoms most often occur in children of mothers without depression who do not accompany children at meals. By providing the appropriate frequencies (only % of YES answers), any doubts will be clarified.
- The discussion is very short and disordered. It gives the impression of a combination of loose paragraphs written by different people. For example, the fourth paragraph on page eight is an injected thread that does not match the previous paragraphs. The discussion would be more structured if it corresponded to the research problems. As a matter of fact, mainly the results from Table 3 were discussed.
Minor comments
- Although the sample size is large, the limitations of the analyses also include obtaining data on less than half of the children from 3,995 pre-selected mother-child pairs.
- Strengths and weaknesses are discussed in two places. I suggest joining and expanding these parts.
- Underneath Table 2 there should be a footnote explaining what “adjusted” means (variables included in the model), and the model’s specification should be identified in the title and footnote of Table 3.
- In this journal, the summary is usually provided in the form of a continuous text in narrative form, with no section titles. It is worth emphasising in the summary that mother-child pairs were investigated.
- The summary features a description of the results from the mother's perspective, not the child's, which stands out unfavourably. It is better to write that the risk of depression increases in children, not that the risk of having a child with depression increases for the mother (which suggests prenatal factors).
- Please check the journal titles in the literature. Occasionally, no capital letters were used, for example, item [24].
Author Response
International Research of Environmental Research and Public Health ijerph-730188R1
The Association of Maternal Accompaniment at Family Dinners and Depressive Symptoms of Korean Adolescents
Dear Reviewer,
I sincerely thank you for your valuable comments and suggestions on our manuscript. We have revised the manuscript accordingly, and detailed revisions were highlighted in yellow in the text and listed below point by point.
Summary of the changes
We reanalyzed the data to apply the reviewer’s opinion, and the number of mother-child pairs were increased from 1,834 to 2,183 in the process. Variables concerning work schedules of the mother, working hours of the mother, and the age groups of adolescents has been added. Many modifications have been made in the discussion and conclusion due to the above changes.
Review
The advantage of the paper is the extensive collected research material covering 1,834 adolescents and their mothers. The tables were also prepared in an extremely careful manner. However, I would like to point out a number of shortcomings in the submitted manuscript. First, you should: indicate the aim of the paper more clearly, complete the description of research tools, thoroughly improve the discussion, including the part describing the strengths and weaknesses of the paper. When discussing the limitations of the study, it would be worthwhile to address the following comments.
Main reservations:
Introduction
- 1. Introduction should end with a clearly defined aim and research questions. Research questions should be consistent with the results presented sequentially and give direction to the discussion.
Reply and correction: Thanks to the comment, we realized that we did not state the aim of the study clearly in the introduction. We added a sentence stating the aim of this study in the last paragraph of our introduction on p.2, which consists of points that are also mentioned in the discussion. The following sentence is what we added:
“Taking the factors mentioned above into consideration, this study aims to analyze the association between maternal accompaniment at family dinners and the depressive symptoms of adolescents.”
- The main result variable identifying children and mothers with depression symptoms is a very simple tool – a single question with Yes / No answer categories. As far as the wording of the question is acceptable, simplified categories of answers seem to be questionable. This should be highlighted as a weakness of the study.
Reply and correction: Thank you for the comment. We added the sentence in the first paragraph of ‘Strength and limitation section of this study’ section on p.10 regarding this point as follows:
“Also, the important variables of adolescent depression and maternal depression were only divided into two categorical variables, “yes/no.” A more thorough analysis would have been possible if these variables had been divided into more categories regarding the severity of depression.”
- The percentage of adolescents with depression symptoms is low. It has not been compared with other studies.
Reply and correction: In previous studies, the rate of depressive symptoms in Korean adolescents was shown as 8.7%, which is in line with our findings, which is 8.0%. The slight difference of the rate of depressive symptoms seems to derive from the fact that this study collected data from identified mother-child pairs.
According to previous study regarding mental health of adolescent, the adolescents who are orphans, or living with their father or grandparents were found to be more likely poor mental health due to experiencing stressful events such as neglect and abuse (Organization, 2014; Räikkönen, Pesonen, Roseboom, & Eriksson, 2012; Walker et al., 2011). In this study, the adolescents who are orphans, or living with only their father or grandparents were not included. This study only included mother-child pairs for the aim of this study. Therefore, we assume that the slight difference of the rate of depressive symptoms came from the different study population.
We also added a sentence to compare the rate of depressive symptoms of adolescents shown in other studies in Korea with the results of this study to the first paragraph of Discussion in p.8:
“The results of this study show that the rate of depressive symptoms in adolescents is 8.0%, which is in line with previous studies that showed that the rate of depressive symptoms in adolescents to be around 8.7% [2, 3]”
- "Behavioural restrictions" turned out to be an important variable. This part of the questionnaire was not described, nor was this significant correlation referred to in the discussion.
Reply and correction: Firstly, we replaced the term “behavioral restriction” to “limitation of activity” to make the meaning of the term clearer. We also added explanation about how we categorized the factors that were unexplained on our first draft. We revised the paragraphs explaining the assessments and categorization of variables to the following paragraphs in the third and fourth paragraphs of 2.3. Measures section in p.4:
“The adolescents’ demographic factors such as their age (12-15); 16 and older) and sex (male; female) were taken into consideration. The adolescents’ dietary condition was categorized as “ample amounts of various foods,” “sufficient food without much variety,” “sometimes do not have enough food,” and “often do not have enough food.” We combined these last two categories into a new group: “insufficient amount of food.” Subjective body cognitions were classified into five groups: “severely skinny,” “slightly thin,” “average,” “slightly obese,” and “severely obese.” We reclassified these categories into “underweight,” “average,” and “overweight.” To assess limitation of activity in both the mother and adolescent, we used their answers to the following question: “Are you currently restricted to your daily life and social activities due to health problems or physical or mental disorders?” We counted affirmative answers as “limited activity” and negative responses as “activity not limited.” The adolescents’ regular walking exercise was assessed based on the frequency of walking, and it was assessed as the number of days walking more than 10 minutes per week, with response options of 0 to 7 days. We reclassified their answers into yes/no.
For maternal demographic factors, maternal age (30-40; 41-45; 46 and older), education level (middle school graduate or less; high school graduate; college graduate or more), household income level (low, middle, and high), and residential area (city; rural) were used. Marriage status of the mother was assessed by the answer to the question, “What is your current marital status?”. The answers were classified into four groups: “married and living together,” “married and living separately,” “separation by death,” and “divorce.”
In addition, we checked the association between limitation of activity and depressive symptoms on children, and how the association between mother accompaniment in family dinners and depressive symptoms on children changes by children’s limitation of activity as you requested. However, we couldn’t find any significant association in neither the result from the main analysis (Table 2) nor the subgroup analysis. Therefore, we decided it would be better not to state about the effect of limitation of activity.
- The age factor has not been considered, and more frequent depressive symptoms can be expected in older adolescents. It has not been examined how the influence of the mother accompaniment in family dinners (and other mother related variables) changes with the child's age.
Reply and correction: Thank you for the comment. We agreed with your comment that the age factor might make a difference, so we analyzed the data with age subgroups: a subgroup with younger adolescents of ages 12-15 and another subgroup of older adolescents of 16 and more. We found out that maternal accompaniment in family dinners and adolescent depression was associated stronger in younger adolescents of ages 12-15, while the association was not as strong in those of ages 16 or older.
Therefore, we added the following sentence to the last paragraph of the result on p.5:
“Finally, the odds of younger adolescents(ages 12-15) having depression were higher when the mother did not eat dinner with them than when they did (OR=1.609, 95% CI: 1.055-2.452), while the older ones(ages 16 and above) did not show significant difference.”
We also added the following sentences in the fourth paragraph of the discussion on p.9:
“Also, the association was strong among the adolescents aged 12-15. According to Hadiwijaya et al., as adolescents get older, hormonal changes act as the main force driving them to separate themselves from their parents to become autonomous and independent individuals [23]. It can be assumed that the importance of maternal accompaniment fades as the adolescents mature and perceive mental independence from their mothers”
- Table 3 should be supplemented with a column with the percentage of children with depression symptoms. The reader may assume (looking only at OR values) that, for example, depression symptoms most often occur in children of mothers without depression who do not accompany children at meals. By providing the appropriate frequencies (only % of YES answers), any doubts will be clarified.
Reply: As you requested, we calculated column percentage of children with depressive symptoms in Table 3 (Column percent (%) based on 174 adolescents the total number of participants with depressive symptoms.)
- The discussion is very short and disordered. It gives the impression of a combination of loose paragraphs written by different people. For example, the fourth paragraph on page eight is an injected thread that does not match the previous paragraphs. The discussion would be more structured if it corresponded to the research problems. As a matter of fact, mainly the results from Table 3 were discussed.
Reply and correction: Thank you for the comment. We eliminated the paragraph that you mentioned, and heavily revised the discussion. We added more discussions about the results from Table 2, and paragraphs discussing results from Table 3 were also revised. We also added a paragraph discussing the political implication for our findings.
The following paragraphs are the ones we added to discuss the results from Table 2 in the 2nd and 3rd paragraphs of the discussion on p. 8:
“Our findings are consistent with previous studies that assess the influence of family dinner on children’s mental health. Elgar et al. demonstrated that the frequency of family dinners is negatively correlated to internalizing and externalizing symptoms and positively correlated to emotional well-being, prosocial behavior, and life satisfaction [11]. According to Musick et al., family dinners may protect children from depression and risky behaviors by providing a regular and comforting context to check in with parents about their day‐to‐day activities and to connect with them emotionally [18].
Our findings, which showed an association between a mother’s absence at family dinners and higher odds of depressive symptoms for her adolescent child, can be explained by possible mechanisms. Links between family dinners and mental health are partially attributable to the ease of communication between adolescents and parents [12]. Levin and Currie examined the association between mother–child communication and life satisfaction among youth, and found that the ease with which they can communicate with their mothers serves as a protective factor for mental well-being [19]. In addition, mother-child communication was found to be a protective factor for the mental status of adolescents, even during stress conditions such as cancer and death of a father [20]. A mothers' warm, sensitive, and engaged communication is associated with lower levels of maladaptive grief and depressive symptoms in her children [20]. As adolescents mature, they generally tend to become oriented toward their peers than their mothers, which leads to the decrease of time for mother-child interaction [21]. Family dinner is proper way to provide adolescents with chances for mother-child conversation, maternal advice giving, and discussions of emotions [22]. Therefore, it is understandable that maternal accompaniment in family dinners have a strong correlation with depressive symptoms in adolescents.”
The following paragraph is what we added to discuss the political implications of this study, on the last paragraph in the discussion on p.9:
There are several political implications for our findings. In order to decrease the possibility of adolescent depression, government policies that emphasize the importance of maternal attendance at family dinners should be reinforced. Such policies should help not only mothers, but society as a whole to acknowledge the importance of maternal attendance at family dinners for adolescent mental health. Mothers, especially those with younger adolescents around the ages 12 to 15, should be encouraged to have dinner with her children through government programs and projects. It is necessary that new policies create an environment where mothers can provide emotional support and engage in daily conversation with her adolescent children through accompanying them at family dinners.”
We also heavily revised the conclusion on p.10 to match our new discussion:
“The findings of this study suggest that maternal accompaniment in family dinners and depressive symptoms in adolescents are strongly associated. In particular, the association between maternal participation in family dinners and an adolescent’s depressive symptoms was most prevalent with those adolescents aged 12-15, female adolescents, those with mothers without depressive symptoms, and city dwellers. In order to decrease the possibility of adolescent depression, it might be necessary to create policies and projects, as well as a social atmosphere, that encourage mothers to participate in family dinners for her adolescent children.”
Minor comments
- Although the sample size is large, the limitations of the analyses also include obtaining data on less than half of the children from 3,995 pre-selected mother-child pairs.
Reply and correction: Thank you for the comment. As we reanalyzed the data, we found out that one of the variables were mistakenly coded, and were able to collect data from 2,183 pairs, which is slightly more than 1,834 pairs that we originally analyzed. However, we agreed to your point. Therefore, regarding that point, we added the following sentence in the first paragraph of the limitation section on p.10:
“The fact that many mother-child pairs were excluded due to the lack of data on considered variables could also be a limitation.”
- Strengths and weaknesses are discussed in two places. I suggest joining and expanding these parts.
Reply: Thank you for the comment. As you recommended, we joined and expanded the parts regarding strengths and weaknesses.
- Underneath Table 2 there should be a footnote explaining what “adjusted” means (variables included in the model), and the model’s specification should be identified in the title and footnote of Table 3.
Reply and correction: Thank you for your comment. First, Table 2 shows the result of logistic regression analysis to find the association between maternal accompaniment in family dinners and depressive symptoms of adolescent after adjusting possible confounders. As you mentioned, we added a footnote regarding the meaning of “adjusted” underneath Table 2, 3 as follows: †Adjusted for offspring's factors(age, sex, limitation of activity, subjective body cognition, and regular walking exercise) and maternal factors(age, level of education, household income level, marriage status, region, work schedule, working hours, depression, and limitation of activity), and year
In addition, the title of Table 3 was modified to: “Table 3. Result of subgroup analysis of the association between maternal accompaniment at family dinners and depressive symptoms by adolescent’s sex, adolescent’s age, maternal depressive symptoms, and region.”
Footnote was also added underneath Table 3 to provide information about the models as follows: “† Stratified Adjusted OR by adolescent’s sex, adolescent’s age, mother's depressive symptoms, and region was calculated by adjusting offspring's factors(age, sex, limitation of activity, subjective body cognition, and regular walking exercise) and maternal factors(age, level of education, household income level, marriage status, region, work schedule, working hours, depression, and limitation of activity), and year after stratifying by adolescent’s sex, adolescent’s age, mother's depressive symptoms, and region.”
- In this journal, the summary is usually provided in the form of a continuous text in narrative form, with no section titles. It is worth emphasizing in the summary that mother-child pairs were investigated.
Reply: Following your advice, we revised the abstract in the form of continuous text in narrative form. We also used the word mother-child pairs several times in the revised abstract.
- The summary features a description of the results from the mother's perspective, not the child's, which stands out unfavourably. It is better to write that the risk of depression increases in children, not that the risk of having a child with depression increases for the mother (which suggests prenatal factors).
Reply and correction: Thank you for the comment. We revised the abstract on p.1 into describing the results from the child’s perspective.
Abstract: This study aims to investigate the association between family dinners involving the participation of both mother and her adolescent child and depressive symptoms within the adolescents. Data from 2,183 mother-child pairs obtained from the Korea National Health and Nutrition Examination Survey IV-VI (2010-2013, and 2015) were employed in the analysis. The dependent variable of this study was depressive symptom of adolescents. Maternal accompaniment at family dinners was the variable of interest. Logistic regression analysis was performed to calculate odds ratios(ORs) and 95% confidence intervals(CI) to analyze the association between family dinners including both mother and adolescent and depressive symptoms within the adolescent. According to the results, maternal absence in family dinners was significantly associated with higher odds of depressive symptoms in adolescents(OR=1.42, 95%CI: 1.01-1.99). In particular, the association was strong among adolescents aged 12-15, female adolescents, those with mothers without depressive symptoms, and city dwellers. This study showed that maternal absence at family dinners was strongly associated with depressive symptoms of adolescents. For the sake of adolescent mental health, it may be necessary to consider the implementation of policies that emphasize the importance of maternal accompaniment at family dinners and encourage the presence of mothers at the dinner table.
- Please check the journal titles in the literature. Occasionally, no capital letters were used, for example, item [24].
Reply and correction: Revised.
References
- Räikkönen, K.; Pesonen, A.-K.; Roseboom, T.J.; Eriksson, J.G. Early determinants of mental health. Best Practice & Research Clinical Endocrinology & Metabolism 2012, 26, 599-611.
- Organization, W.H. Violence and injury prevention-adverse childhood experiences international questionnaire (ace-iq). Available online: https://www.who.int/violence_injury_prevention/violence/activities/adverse_childhood_experiences/en/ (accessed on 25 Feb 2020).
- Walker, S.P.; Wachs, T.D.; Grantham-McGregor, S.; Black, M.M.; Nelson, C.A.; Huffman, S.L.; Baker-Henningham, H.; Chang, S.M.; Hamadani, J.D.; Lozoff, B. Inequality in early childhood: Risk and protective factors for early child development. The lancet 2011, 378, 1325-1338.
We would like to thank you for your consideration of our revised manuscript for publication. We look forward to a positive response.
Sincerely,
Corresponding Author of ijerph-730188
Reviewer 2 Report
The manuscript titled "The Association of Maternal Accompaniment in
Family Dinners and Depressive Symptoms of Korean
Children and Adolescents" aims to investigate the association of the participation of mothers with children in family dinners and the occurrence of depression in those children.
Introduction:
Authors introduce the theme depression but refer to a statistic from United State - this study is though placed in Korea so that those statistics about children and depression should be presented for this country.
In abstract is said that the goal is to investigate the depression association for mothers with children but in Introduction (second paragrah) authors declare that they aim to investigate the depression among adolescents.
Further in other paragraphs the term children and adolescents are mixed and it is not clear to which group the study is aimed for, since authors presented different statistics for children and adolescents regarding frequency of events with depression, this term should be better used and explained.
Materials and Methods:
Participant selection, randomization and allocation were good described.
Instruments and variables used in the research that are important for understanding the aim of this study were clearly described.
Statistical methods used are ok.
Results:
Table 1: Why is only the overall mean for age presented? This sould also be presented for both groups (no depressive / depressive) as for the other variables.
Authors ground their findings and argumentation on Odds Ratios and only speaking about a greater OR for a group doesn't mean automatically that this difference is statistical significant, maybe a further analysis like hypothesis testing would deliver clearer results.
Discussion:
Authors conclude and discuss their hypothesis based on the results. Association does not mean causation and it fells like the association described and founded in the analysis was assumed to be the final causation for depression in participants, be carefull when concluding such analysis.
The study could be better analyzed, there is a good amount of information but the methods used are unsufficient to deliver the final conclusion.
Author Response
International Research of Environmental Research and Public Health ijerph-730188R1
The Association of Maternal Accompaniment at Family Dinners and Depressive Symptoms of Korean Adolescents
Dear Reviewer,
I sincerely thank you for your valuable comments and suggestions on our manuscript. We have revised the manuscript accordingly, and detailed revisions were highlighted in yellow in the text and listed below point by point.
Summary of changes
Variables concerning work schedules of the mother, working hours of the mother, and the age groups of adolescents has been added. The term ‘children’ has been eliminated, and subgroup analysis regarding the age group (12-15, 16 or above) of adolescents has been done. Many modifications have been made in the discussion and conclusion due to the above changes.
Review
The manuscript titled "The Association of Maternal Accompaniment in Family Dinners and Depressive Symptoms of Korean Children and Adolescents" aims to investigate the association of the participation of mothers with children in family dinners and the occurrence of depression in those children.
Introduction:
- Authors introduce the theme depression but refer to a statistic from United State - this study is though placed in Korea so that those statistics about children and depression should be presented for this country.
Reply and correction: Thank you for the comment. We replaced the statistic from United States to a statistic from South Korea. We added the following sentences to the first paragraph of the introduction on p.1:
“According to studies by Ma and Cho and Kim et al., approximately 8.7% of adolescents in the Republic of Korea are identified as depressed [4,5]. The high rates of depressive symptoms in Korean adolescents underscore the importance of investigating the factors related to adolescent depression.”
- In abstract is said that the goal is to investigate the depression association for mothers with children but in Introduction (second paragrah) authors declare that they aim to investigate the depression among adolescents.
Reply and correction: Our intention was to show the importance of investigating factors related to depressive symptoms of adolescents, and then introduce our point about family dinners. We realized the lack of explanation of our intentions, therefore added the following sentence at the end of the first paragraph of the introduction on p.1:
“The high rates of depressive symptoms in Korean adolescents underscore the importance of investigating the factors related to adolescent depression.”
- Further in other paragraphs the term children and adolescents are mixed and it is not clear to which group the study is aimed for, since authors presented different statistics for children and adolescents regarding frequency of events with depression, this term should be better used and explained.
Reply and correction: Thank you for the comment. The statistics that are mentioned has been replaced by another statistic from a study in Korea. We realized that we have been using the term “children” and “adolescents” confusingly, so we unified the terms to “adolescents”, and used the term “children” only when using the phrase “the mother and her children.” The participants of this study are mother-child pairs, in which the child’s age is between 12 to 19. Therefore, we decided that the term “adolescent” is adequate to be used as a term to name those participants.
Materials and Methods:
- Participant selection, randomization and allocation were good described.
Instruments and variables used in the research that are important for understanding the aim of this study were clearly described.
Statistical methods used are ok.
Reply: Thank you for the comment.
Results:
- Table 1: Why is only the overall mean for age presented? This could also be presented for both groups (no depressive / depressive) as for the other variables.
Reply: We apologize for our mistake. The p-value came from T-test for comparison of mean of age by having depressive symptoms.
1) Adolescent Age (Mean±S.D.)
Overall adolescent mean was 14.5 ± 1.9. Those adolescents with depressive symptoms showed older age (those with no depressive symptoms: 14.4 ± 1.9; those with depressive symptoms: 14.6 ± 1.9), but the difference was not statistically significant (p-value: 0.2334).
2) Maternal age (Mean±S.D.)
Overall maternal mean age was 43.2 ± 4.2. Those mothers without depressive symptoms showed older age (those with no depressive symptoms: 43.2 ± 4.1; those with depressive symptoms: 42.9 ± 4.5), but the difference was not statistically significant (p-value: 0.397).
In this study, we used categorized age for both adolescents (12-15; 16 and older) and mothers (30-40; 41-45; 45 and older) to analyze. Those who having depressive symptoms was shown to be 7.8% and 8.4% in 12-15 and 16 and older age group of adolescents, respectively (P = 0.639). Those who having depressive symptoms was shown to be 9.1%, 7.4%, and 7.9% in 30-40, 41-45, and 46 and older age group of mothers, respectively (P = 0.476). To provide information for readers, we also added overall mean for age and mean age by having depressive symptoms is represented in Table 1.
- Authors ground their findings and argumentation on Odds Ratios and only speaking about a greater OR for a group doesn't mean automatically that this difference is statistically significant, maybe a further analysis like hypothesis testing would deliver clearer results.
Reply and correction: Thank you for the comment. We realized that we made a mistake of introducing insignificant data without noting that they were insignificant. We revised the third paragraph of the result section on p.5 as follows:
“Table 3 shows adjusted ORs of depression in adolescents according to whether mothers ate dinner with their children, stratified by adolescent’s sex, residential area, and whether the mother has depression. For the adolescent’s sex stratification, we found that the odds of a girl having depression were higher when her mother was not present at dinner than when her mother was present at dinner (OR = 1.814, 95% CI: 1.085-3.032). There was no significant difference between the odds of experiencing depressive symptoms for adolescent males who ate with their mother and those who did not (OR = 1.243, 95% CI: 0.763-2.025). Regarding region, adolescents who lived in a city had higher odds of suffering depressive symptoms when their mother was absent at family dinners (OR=1.724, 95% CI: 1.167-2.548), while the odds of adolescents who live in rural areas were not significantly different. For mother with depression, we found higher odds of depression in adolescents whose mother had no depressive symptoms and did not eat dinner with them (OR=1.92, 95% CI=1.277-2.888), while adolescents with mothers who had depressive symptoms showed no significant values. Finally, the odds of younger adolescents (ages 12-15) having depression were higher when the mother did not eat dinner with them than when they did (OR=1.609, 95% CI: 1.055-2.452), while the older ones (ages 16 and above) did not show significant difference.”
Discussion:
- Authors conclude and discuss their hypothesis based on the results. Association does not mean causation and it feels like the association described and founded in the analysis was assumed to be the final causation for depression in participants, be careful when concluding such analysis.
Reply and correction: Thank you for the comment. We replaced the terms “effect” to “correlation,” the phrase “has a positive effect on” to “has significant correlation with.”, and “had more impact on” to “had a stronger association with.” The following sentences are what we revised:
“The study results showed that the absence of the mother during dinner was significantly associated with higher odds of depressive symptoms in adolescents.” (1st paragraph of discussion, p.8)
“Our findings, which showed an association between a mother’s absence at family dinners and higher odds of depressive symptoms for her adolescent child, can be explained by possible mechanisms.” (3rd paragraph of discussion, p.8)
- The study could be better analyzed, there is a good amount of information but the methods used are insufficient to deliver the final conclusion.
Reply: We understand that our former method and result were not sufficient to deliver our former conclusion. We feel sorry for the former mistake. The aim of this study is to identify the association of maternal accompaniment at family dinners and depressive symptoms of Korean adolescents. Previously, we tried to explain the association between the adolescents having family dinner not accompanied by mother and higher depressive symptoms of adolescents by the possible effect from the poor work environment of mothers (ex. long work hours, night work) in texts. However, there was a lack of analysis. To solve this problem, we make some changes in analysis.
First, we tried to include more useful information including age of adolescent child, working schedule of mothers, and working hours of mothers. Especially, a reason for adding work related variables (ex. working schedule of mothers, working hours of mothers) is to consider possible effect of these variables on the association between maternal accompaniment at family dinners and adolescents’ depressive symptoms. Therefore, these variables were additionally added in our model. When we checked the adjusted odds ratios for the association between maternal accompaniment in family dinners and depressive symptoms of adolescent, we found that the adolescents having family dinner not accompanied by mother more likely associated with higher odds of depressive symptoms. However, there were no statistically significant association found between work related variables (ex. working schedule of mothers, working hours of mothers) and depressive symptoms. In addition, we tried to seek the different association between maternal accompaniment at family dinners and adolescents’ depressive symptoms by subgroup analysis stratified to working schedule of mothers, working hours. However, no significant association was found. You can check the result of the subgroup analysis stratified to working schedule of mothers, working hours from Table A below:
Table A. Result of subgroup analysis in the association between maternal accompaniment in family dinners and depressive symptoms by working schedule of mothers, working hours
|
Variable |
Child depression |
||||
|
OR |
95% CI |
p-value |
|||
|
Work schedule |
|||||
|
Day work |
Not accompanied by mother/child |
1.470 |
0.935 |
2.309 |
0.095 |
|
Accompanied by mother/child |
1.000 |
||||
|
Night work |
Not accompanied by mother/child |
1.731 |
0.350 |
8.567 |
0.502 |
|
Accompanied by mother/child |
1.000 |
||||
|
Shift work and others |
Not accompanied by mother/child |
<0.001 |
<0.001 |
>999.999 |
0.625 |
|
Accompanied by mother/child |
1.000 |
||||
|
Unemployed |
Not accompanied by mother/child |
1.266 |
0.650 |
2.466 |
0.488 |
|
Accompanied by mother/child |
1.000 |
||||
|
Working hour |
|||||
|
0-20 |
Not accompanied by mother/child |
1.200 |
0.695 |
2.072 |
0.971 |
|
Accompanied by mother/child |
1.000 |
||||
|
21-40 |
Not accompanied by mother/child |
1.916 |
0.937 |
3.915 |
0.075 |
|
Accompanied by mother/child |
1.000 |
||||
|
40-51 |
Not accompanied by mother/child |
1.211 |
0.463 |
3.169 |
0.696 |
|
Accompanied by mother/child |
1.000 |
||||
|
52 and over |
Not accompanied by mother/child |
1.323 |
0.446 |
3.920 |
0.614 |
|
|
Accompanied by mother/child |
1.000 |
|
|
|
Despite several limitations, this study found some meaningful results. The absence of the mother during dinner was significantly associated with higher odds of depressive symptoms in adolescents. In particular, the association between maternal accompaniment at family dinners and an adolescents’ depressive symptoms was most strongly prevalent among adolescents aged 12-15, female adolescents, those with mother’s without depressive symptoms, and city dwellers. This result gives directions to create policies and projects to decrease the possibility of adolescent depression by encourage mothers to participate in family dinners. We believe that we tried our best to use proper methods, and the revised conclusion of this study matches the results of analysis compared to the former sentences. We also revised sentences regarding this issue in Method, Discussion, and Conclusion section. We apologize for this mistake again. We would like to request your kind understanding.
We would like to thank you for your consideration of our revised manuscript for publication. We look forward to a positive response.
Sincerely,
Corresponding Author of ijerph-730188

Reviewer 3 Report
Dear Authors,
The study is well-designed, large sampled, analyzing mother and child matched cross-sectional relations between family dinner and child depression. No study hypotheses are presented in the background.
However, study methods are not described well enough.
The main limitation is understudied major study variable, depressive symptoms. A single question was used for the adults and adolescents of the very broad age range. The authors do not provide evidence about the validity and sensitivity of using this single question. This limitation remains undiscussed.
Next, the authors provide many confounding variables that are undescribed in the methods section. For example, economic activity, behavior restriction, marriage status, residential area, and income are not described in the methods section, so it remains unclear, how these variables were measured.
It would be very helpful to provide p values together with the odds ratios rather than go through CIs considering whether this difference was significant or not.
Table 1. If the average age is presented for the whole sample (children and mothers) without a comparison between depressive symptoms and no depressive symptoms groups, it is not clear where the p value comes from?
Please reconsider leaving one decimal place in Table 1 and two decimal places in Tables 2 and 3.
Author Response
International Research of Environmental Research and Public Health ijerph-730188R1
The Association of Maternal Accompaniment at Family Dinners and Depressive Symptoms of Korean Adolescents
Dear Reviewer,
I sincerely thank you for your valuable comments and suggestions on our manuscript. We have revised the manuscript accordingly, and detailed revisions were highlighted in yellow in the text and listed below point by point.
Summary of changes
Variables concerning work schedules of the mother, working hours of the mother, and the age groups of adolescents has been added. The term ‘children’ has been eliminated, and subgroup analysis regarding the age group (12-15, 16 or above) of adolescents has been done. Several limitations have been added to the ‘Strength and limitation’ section to apply the reviewer’s opinions. Many modifications have been made in the discussion and conclusion due to the above changes.
Review
The study is well-designed, large sampled, analyzing mother and child matched cross-sectional relations between family dinner and child depression. No study hypotheses are presented in the background.
However, study methods are not described well enough.
- The main limitation is understudied major study variable, depressive symptoms. A single question was used for the adults and adolescents of the very broad age range. The authors do not provide evidence about the validity and sensitivity of using this single question. This limitation remains undiscussed.
Reply and correction: Thank you for the comment. We added the sentence regarding this point in the first paragraph of the ‘Strengths and limitations of this study’ section on p.10 as follows:
“Also, the important variables of adolescent depression and maternal depression were only divided into two categorical variables, “yes/no”. A more thorough analysis would have been possible if these variables had been divided into more categories regarding the severity of depression.”
- Next, the authors provide many confounding variables that are undescribed in the methods section. For example, economic activity, behavior restriction, marriage status, residential area, and income are not described in the methods section, so it remains unclear, how these variables were measured.
Reply and correction: We replaced the term “behavioral restriction” to “limitation of activity” to make the meaning of the term clearer. We also added explanation about how we categorized the factors that were unexplained on our first draft. We heavily revised the method section to clarify how the variables were measured. The following paragraphs are the ones we revised in the third and fourth paragraphs of 2.3. Measures on p. 4:
“The adolescents’ demographic factors such as their age (12-15); 16 and older) and sex (male; female) were taken into consideration. The adolescents’ dietary condition was categorized as “ample amounts of various foods,” “sufficient food without much variety,” “sometimes do not have enough food,” and “often do not have enough food.” We combined these last two categories into a new group: “insufficient amount of food.” Subjective body cognitions were classified into five groups: “severely skinny,” “slightly thin,” “average,” “slightly obese,” and “severely obese.” We reclassified these categories into “underweight,” “average,” and “overweight.” To assess limitation of activity in both the mother and adolescent, we used their answers to the following question: “Are you currently restricted to your daily life and social activities due to health problems or physical or mental disorders?” We counted affirmative answers as “limited activity” and negative responses as “activity not limited.” The adolescents’ regular walking exercise was assessed based on the frequency of walking, and it was assessed as the number of days walking more than 10 minutes per week, with response options of 0 to 7 days. We reclassified their answers into yes/no.
For maternal demographic factors, maternal age (30-40; 41-45; 46 and older), education level (middle school graduate or less; high school graduate; college graduate or more), household income level (low, middle, and high), and residential area (city; rural) were used. Marriage status of the mother was assessed by the answer to the question, “What is your current marital status?”. The answers were classified into four groups: “married and living together,” “married and living separately,” “separation by death,” and “divorce.”
- It would be very helpful to provide p values together with the odds ratios rather than go through CIs considering whether this difference was significant or not.
Reply: As you requested, we represented p values together in Table 2, 3.
- Table 1. If the average age is presented for the whole sample (children and mothers) without a comparison between depressive symptoms and no depressive symptoms groups, it is not clear where the p value comes from?
Reply: We apologize for our mistake. The p-value came from T-test for comparison of mean of age by having depressive symptoms.
1) Adolescent Age (Mean±S.D.)
Overall adolescent mean was 14.5 ± 1.9. Those adolescents with depressive symptoms showed older age (those with no depressive symptoms: 14.4 ± 1.9; those with depressive symptoms: 14.6 ± 1.9), but the difference was not statistically significant (p-value: 0.2334).
2) Maternal age (Mean±S.D.)
Overall maternal mean age was 43.2 ± 4.2. Those mothers without depressive symptoms showed older age (those with no depressive symptoms: 43.2 ± 4.1; those with depressive symptoms: 42.9 ± 4.5), but the difference was not statistically significant (p-value: 0.397).
To provide information for readers, we added overall mean for age and mean age by having depressive symptoms is represented in Table 1.
- Please reconsider leaving one decimal place in Table 1 and two decimal places in Tables 2 and 3.
Reply: As you requested, we tried to display one decimal place in Table 1 and two decimal places in Tables 2 and Tables 3 except for p-value with three decimal places.
We would like to thank you for your consideration of our revised manuscript for publication. We look forward to a positive response.
Sincerely,
Corresponding Author of ijerph-730188R1

Round 2
Reviewer 1 Report
Answers to the reviewer's comments are very accurate. The authors also amended the text accordingly.
Author Response
International Research of Environmental Research and Public Health ijerph-730188R2
The Association of Maternal Accompaniment at Family Dinners and Depressive Symptoms of Korean Adolescents
Dear Reviewer,
I sincerely thank you for your previous valuable suggestions on our manuscript. Our manuscript improved due to your suggestions. Thank you so much again.
Review
Answers to the reviewer's comments are very accurate. The authors also amended the text accordingly.
Reply: I do really appreciate you.
We would like to thank you for your consideration of our revised manuscript for publication.
Sincerely,
Corresponding Author of ijerph-730188R2

Reviewer 2 Report
Authors have either added new sections to better explain the aim and hypothesis of this study or have changed the sections where any unclear statements could be found.
After this revision it seems ok to be published but still lack on detailed analysis.
Author Response
International Research of Environmental Research and Public Health ijerph-730188R2
The Association of Maternal Accompaniment at Family Dinners and Depressive Symptoms of Korean Adolescents
Dear Reviewer,
I sincerely thank you for your valuable comments and suggestions on our manuscript. We have revised the manuscript accordingly, and detailed revisions were highlighted in yellow in the text and listed below point by point.
Review
Authors have either added new sections to better explain the aim and hypothesis of this study or have changed the sections where any unclear statements could be found.
After this revision it seems ok to be published but still lack on detailed analysis.
Reply: Thank you for the comment. The aim of this study was to investigate the association between family dinners involving the participation of both mother and her adolescent child and depressive symptoms within the adolescents. To achieve that goal, we tried to control more variables than previous studies in order to identify the association. Furthermore, variables such as working hours and work schedule of the mother, as well as the age groups of the adolescent, were added to develop our research during the previous revision. Through this process, this study showed that maternal absence at family dinners was strongly associated with depressive symptoms of adolescents. By subgroup analysis, the fact that the association in strong among adolescents aged 12-15, female adolescents, those with mothers without depressive symptoms, and city dwellers was also brought to light. Although there are many limitations, we believe this study is meaningful, especially since depressive symptoms in adolescents are a serious social issue. We would like to request your kind understanding.
We would like to thank you for your consideration of our revised manuscript for publication. We look forward to a positive response.
Sincerely,
Corresponding Author of ijerph-730188R2

Reviewer 3 Report
Dear Authors,
Thank you for the improved version of the manuscript.
It would be worth to add to the limitations section, that not only dichotomized answers of the major study variable is a limitation. The use of the depression scale for adolescents and their mothers would demonstrate a deeper examination of the main variable.
Author Response
International Research of Environmental Research and Public Health ijerph-730188R2
The Association of Maternal Accompaniment at Family Dinners and Depressive Symptoms of Korean Adolescents
Dear Reviewer,
I sincerely thank you for your valuable comments and suggestions on our manuscript. We have revised the manuscript accordingly, and detailed revisions were highlighted in yellow in the text and listed below point by point.
Review
Dear Authors,
Thank you for the improved version of the manuscript.
It would be worth to add to the limitations section, that not only dichotomized answers of the major study variable is a limitation. The use of the depression scale for adolescents and their mothers would demonstrate a deeper examination of the main variable.
Reply and correction: Thank you for the comment. We added the following sentence to explain that point in the first paragraph of the ‘Strengths and limitations of this study’ section on p. 10:
“A deeper examination of the main variable, as well as a more thorough analysis, would have been possible if depression scales for adolescents and their mothers had been used.”
We would like to thank you for your consideration of our revised manuscript for publication. We look forward to a positive response.
Sincerely,
Corresponding Author of ijerph-730188R2
